# S-Nitrosoglutathione Reductase—The Master Regulator of Protein S-Nitrosation in Plant NO Signaling

**DOI:** 10.3390/plants8020048

**Published:** 2019-02-21

**Authors:** Jana Jahnová, Lenka Luhová, Marek Petřivalský

**Affiliations:** Department of Biochemistry, Faculty of Science, Palacky University, Šlechtitelů 11, 78371 Olomouc, Czech Republic; jana.jahnova@upol.cz (J.J.); lenka.luhova@upol.cz (L.L.)

**Keywords:** S-nitrosation, S-nitrosothiols, nitric oxide, S-nitrosoglutathione reductase, S-(hydroxymethyl)glutathione

## Abstract

S-nitrosation has been recognized as an important mechanism of protein posttranslational regulations, based on the attachment of a nitroso group to cysteine thiols. Reversible S-nitrosation, similarly to other redox-base modifications of protein thiols, has a profound effect on protein structure and activity and is considered as a convergence of signaling pathways of reactive nitrogen and oxygen species. In plant, S-nitrosation is involved in a wide array of cellular processes during normal development and stress responses. This review summarizes current knowledge on S-nitrosoglutathione reductase (GSNOR), a key enzyme which regulates intracellular levels of S-nitrosoglutathione (GSNO) and indirectly also of protein S-nitrosothiols. GSNOR functions are mediated by its enzymatic activity, which catalyzes irreversible GSNO conversion to oxidized glutathione within the cellular catabolism of nitric oxide. GSNOR is involved in the maintenance of balanced levels of reactive nitrogen species and in the control of cellular redox state. Multiple functions of GSNOR in plant development via NO-dependent and -independent signaling mechanisms and in plant defense responses to abiotic and biotic stress conditions have been uncovered. Extensive studies of plants with down- and upregulated GSNOR, together with application of transcriptomics and proteomics approaches, seem promising for new insights into plant S-nitrosothiol metabolism and its regulation.

## 1. Introduction

Nitric oxide (NO) is an important messenger included in many physiological processes. It is an uncharged, relatively stable free radical with unpaired electrons allowing diverse chemistry. Rapid reactions with other radicals including reactive oxygen species (ROSs) [1] lead to the formation of reactive nitrogen species (RNSs), substances with versatile chemical properties triggering specific physiological responses. NO is involved in the regulation of plant growth and development, immunity and environmental interactions with the inclusion of signaling cascades of responses to stress conditions [2,3].

In general, the biosynthesis of NO in plants can proceed by pathways either oxidative or reductive, either enzymatic or non-enzymatic reactions, anyway depending on the site and the nature of stimulus for NO production [3,4,5]. The nitrate reductase (NR; EC 1.6.6.1) pathway, localized in the cytosol, is the best-characterized production pathway of NO in plants [4]. Another well-described way of NO production is the nitrite reduction in electron transport chains of mitochondria or chloroplasts [5]. In mammals, constitutive and inducible isoforms of nitric oxide synthase (NOS, 1.14.13.39) are the major enzyme sources of NO. NOS-like enzymatic activities were described in plants, however neither gene with significant homology nor protein with similarity to bacterial or animal NOS have been found [6,7,8,9]. It is assumed that NOS-like activity in plants is carried out by several enzymes, which can together generate NO from L-arginine and have the same cofactors requirements as the NOS in mammals and bacteria [6,7]. However, a recent analysis of 1087 land plant transcriptomes confirmed the absence of evolutionarily conserved NOS sequences within the plant kingdom [9].

S-nitrosothiols (SNOs) represent relatively a stable reserve and transport form of NO in vivo [10,11]. They are formed by S-nitrosation, a selective and reversible covalent addition of nitric oxide moiety to the sulfur atom of cysteines both in low-molecular weight thiols and proteins. S-nitrosation is considered to be an important redox-based post-translational protein modification, an integral part of signaling pathways of NO and RNSs [12]. It is supposed to be implicated in the regulation of a variety of protein functions and cell activities—programmed cell death, metabolism, control of redox balance, iron homeostasis, control of protein quality, and gene transcription [13]. Importantly, S-nitrosothiols are considered key elements of the interplay between RNSs and ROSs, both under physiological and stress conditions leading to various scenarios of oxidative, nitrosative, or nitro-oxidative stress [14].

The most abundant low-molecular weight S-nitrosothiol is suggested to be S-nitrosoglutathione (GSNO), generated by an O_2_-dependent reaction of NO-derived RNSs and the major antioxidant tripeptide glutathione (GSH; γ-Glu-Cys-Gly). GSNO is regarded to be an intracellular reservoir of NO bioactivity and a transport form of NO as well, even though NO and GSNO do not always interact with the same target proteins [15]. Reactions including S-nitrosation, transnitrosation, when nitroso group is transferred from SNO to the thiol group of another molecule, S-glutathionylation are involved in its metabolism [16,17]. Acting as a buffer for NO, GSNO could maintain the level of protein S-nitrosation [18]. However, more detailed knowledge on the distribution, intracellular levels, and modulation of GSNO under natural and stress conditions is needed [17].

## 2. S-Nitrosoglutathione Reductase: Key Enzyme of the Regulation of S-Nitrosation and Formaldehyde Detoxification

GSNOR is an evolutionarily conserved, cytosolic enzyme that catalyzes the NADH-dependent reduction of GSNO, leading to the formation of glutathione disulfide (GSSG) and ammonium [18]. Sakamoto et al. [19] demonstrated for the first time in plants that GSNOR is glutathione-dependent formaldehyde dehydrogenase (FALDH; EC 1.2.1.1). The proper substrate for FALDH is the hemithioacetal S-hydroxymethylglutathione (HMGSH), formed nonenzymically from formaldehyde and glutathione [20]. HMGSH is oxidized to S-formylglutathione using NAD^+^ as a coenzyme (Figure 1A). After the elucidation of an exact reaction mechanism, the enzyme was reclassified as S-(hydroxymethyl)glutathione dehydrogenase (EC 1.1.1.284). Formerly, Koivusalo et al. [21] reported evidence that FALDH and ADH3 are identical enzymes. Thus, in accordance with the formal enzyme classification, GSNOR is a Zn-dependent medium-chain class III alcohol dehydrogenase (ADH3; EC 1.1.1.1). Since the GSNO has been uncovered as among the most effective substrates for this enzyme [18,22,23], the designation as GSNOR is currently widely extended within the scientific literature. However, the denomination has not been accepted by IUBMB nomenclature commission up to the present day.

Via removing GSNO, GSNOR plays a critical role in the metabolism of RNSs, in the homeostasis of intracellular levels of NO and in control of the trans-nitrosation equilibrium between S-nitrosylated proteins and GSNO, the most common low-molecular weight S-nitrosothiol [15,18,22,24]. In trans-nitrosation reactions, the nitroso group is transferred among thiols on proteins and low-molecular weight peptides. GSNO reduction by GSNOR is an irreversible reaction, and the products can no longer nitrosate cellular proteins.

Through the GSNO reductase activity, GSNO is reduced to an unstable intermediate N-hydroxysulfinamide (GSNHOH) in the first reaction step, using NADH as a specific co-substrate (Figure 1B). Different final products are produced in the next reaction step, depending on the local concentration of GSH. Thus, the cellular redox potential in terms of NADH and GSH levels is an important factor in control of the product formation [25]. Common cellular concentrations of GSH are found in a millimolar range, which favors a reaction shift from GSNHOH to the formation of glutathione disulfide (GSSG) accompanied with the release of hydroxylamine [23,25]. However, the cellular levels of GSH are widely fluctuating under different biotic and abiotic stress conditions. In vitro studies have demonstrated that, at low levels of GSH, GSNHOH spontaneously converts to glutathione sulfinamide (GSONH_2_). GSONH_2_ is further hydrolyzed to glutathione sulfinic acid (GSO_2_H), which can be oxidized even to glutathione sulfonic acid (GSO_3_H) under oxidative stress induced by various stress conditions [25]. The latter metabolites inhibit glutathione transferases, enzymes with an important role in the glutathione-dependent detoxification of xenobiotics by their conjugation with GSH [26].

Another factor involved in the regulation of GSNO turnover is the accessibility of NADH, a co-substrate in the reduction of GSNO. The cellular ratio of free oxidized and reduced form of dinucleotides (NAD^+^/NADH) is high under physiological conditions, which is not favorable for reductive pathways [27]. The NADP^+^/NADPH ratio is much lower, which enables NADPH to be used in biosynthetic reductive pathways [28]. Since GSNOR cannot use NADPH in the reduction of GSNO, it is controlled by NADH availability and increasing levels of NADH are proposed to trigger the GSNO reduction. GSNOR enzymes themselves produce NADH in the process of the oxidation of formaldehyde; formaldehyde likely triggers the reduction of GSNO [25].

In plants, formaldehyde can originate from various processes. Among them, the major sources of formaldehyde include the dissociation of 5,10-methylene-tetrahydrofolate and the oxidation of methanol formed by demethylation of pectin. Formaldehyde can also be formed by oxidative demethylation reactions, decarboxylation of glyoxylate, and P450-dependent oxidation of xenobiotics [29,30]. The compound is highly reactive because of the polarized carbonyl group and can participate in a nucleophile as well as an electrophile addition and substitution reactions. The carbonyl group can react with DNA and proteins producing stable carboxylated products. GSNOR, by another name FALDH, is the main enzyme metabolizing formaldehyde [23]. This is implicated in this process by oxidation of HMGSH, spontaneously formed from formaldehyde and glutathione. Emergent S-formylglutathione is decomposed by S-formylglutathione hydrolase (EC 3.1.2.12) to glutathione and formate [29].

## 3. Molecular Properties of S-Nitrosoglutathione Reductase

### 3.1. GSNOR Structure

A few studies on kinetic and structural analysis of plant GSNOR enzymes indicate a high similarity between the plant and human homologues [31,32,33,34,35]. GSNOR described in tomato (*Solanum lycopersicum*; SlGSNOR) plants is a homodimeric enzyme consisting of two 40 kDa subunits containing a big catalytic and a small coenzyme-binding domain with an active site localized in a cleft between them [34]. Non-catalytic domain includes a binding site for NAD^+^ coenzyme; six beta-strands of each coenzyme-binding domain form 12 pseudo-continuous beta-sheets. Each catalytic domain includes two zinc atoms. One of them is involved in the catalytic mechanism by activating the hydroxyl and carbonyl groups of substrates for transfer of hydride, and is bonded to Cys47, Cys177, His69, and either Glu70 or a water molecule. The second zinc atom is considered to have purely a structural role and is coordinated to four cysteine residues, Cys99, Cys102, Cys105, and Cys113 [34].

Crystal structures of SlGSNOR apoenzyme, binary complex with NAD^+^ and a structure crystallized in the presence of NADH and GSH were described to understand the role of specific residues in the active site and the structural changes occurring during the catalytic cycle of GSNOR activity [34]. Catalytic domains of the apoenzyme and of the binary complex with NAD^+^ are both in the semi-open conformation. The catalytic zinc atoms in the apoenzyme are in a tetrahedral configuration, H-bonded to Cys47, Cys177, His69 and coordinated to the molecule of water in the active site. The coenzyme binding is associated with the catalytic zinc atoms movement towards Glu70 in the catalytic domain in a hydrogen-bonding interaction with the carboxylate oxygen of Glu70. Zinc atoms are in a tetrahedral configuration coordinated with Cys47, Cys177, His69, and Glu70, and they are no longer coordinated with the water molecule. In the SlGSNOR structure crystallized with NADH and GSH, the enzyme appears in closed conformation; rotation of the catalytic domains by approximately 3° towards the coenzyme-binding domains was observed [34]. This structure is highly similar to the complex of human GSNOR (hGSNOR) with NADH and HMGSH, where a catalytic domain moves towards the coenzyme-binding domain during the formation of the ternary complex [32,33,36]. In the hGSNOR, the domain closure brings one molecule of water close to 2′-hydroxyl of nicotinamide riboside moiety, suggesting that the proton from the substrate is transferred to the solvent directly from the coenzyme. Similarly, in SlGSNOR, the domain closure brings Thr49 and His48 closer to the 2′-hydroxyl of nicotinamide riboside moiety, which might facilitate the proton transfer [34]. In the hGSNOR, the HMGSH substrate is directly coordinated to active site zinc atoms and interacts with highly conserved residues Arg114, Asp55, Glu57, and Thr46, and the zinc atom is in a tetrahedral configuration coordinated with Cys44, Cys177, His66, and HMGSH [32,33].

Eukaryotic GSNORs are highly conserved and unusually cysteine-rich proteins [35]. Most of the cysteines are inaccessible to the solvent, having usually only a structural function [37]. Three positionally conserved cysteines accessible to the solvent are predicted to be the site of post-translational modifications, e.g., S-nitrosation or glutathionylation [35]. Regulation of GSNOR activity through S-nitrosation of that conserved cysteines was observed in *A. thaliana* plants [15,38]. In vitro studies showed susceptibility of the enzymatic activity to NO donors and its subsequent restoration after treatment with dithiothreitol (DTT), a reducing agent [38]. Mono-, di-, and trinitrosation, which were confirmed by mass spectrometry, lead to subtle changes in enzyme conformation. GSNOR monomers within the same dimer interact with each other and the substrate binding cleft alters the shape. Thus, GSNOR activity might be regulated by high levels of NO donors.

### 3.2. GSNOR Substrate Specificity and Inhibition

Enzymes from the alcohol dehydrogenase class I (ADH1) and class III (ADH3) families have a very similar tertiary structure, but despite this fact their substrate specificity and kinetic mechanism are very different [39,40]. GSNOR, belonging to the class III family, can work in two modes catalyzing a conversion of plenty of substrates, including long-chain primary alcohols, aldehydes, and ω-hydroxyfatty acids. In the dehydrogenase mode, it catalyzes oxidation in the presence of NAD^+^, whereas in the reductase mode it catalyzes reduction in the presence of NADH. NADP^+^ and NADPH are very poor coenzymes reaching negligible reaction rates compared to those with NAD^+^ and NADH [34].

Several studies on hGSNOR showed that an anion binding pocket, containing Gln111, Arg114, and Lys283, is presented in the active site of hGSNOR, and the positive charge of Arg114 enables correct orientation of negatively charged substrates, HMGSH and GSNO [26,32,33,36]. The plant GSNOR enzyme exhibits significant difference in the anion-binding pocket of the active site, which is composed of only two residues, Arg117 and Lys287, while the glutamine (Gln111 in hGSNOR) is missing and replaced by Gly114. Since the Gln111 in hGSNOR forms a hydrogen bond with carboxyl oxygen atoms of substrate, the different composition of the anion binding pocket of plant GSNOR results in the reduced affinity for the carboxyl group of ω-hydroxyfatty acids [34].

Plant GSNOR catalyzes the oxidation of HMGSH, geraniol, cinnamyl alcohol, ω-hydroxyfatty acids, and aliphatic alcohols with chains longer than four carbons, to corresponding aldehydes using NAD^+^ as a coenzyme. Short-chain alcohols, e.g., ethanol and propanol, are not enzyme substrates. In the reductase mode, plant GSNOR preferentially catalyzes the reduction of GSNO, while reactions with either aliphatic or aromatic aldehydes are insignificant. The observed K_m_ values for various plausible substrates of SlGSNOR were in the same range as those for AtGSNO, which indicates that the substrate preferences of plant GSNOR are similar [34,41]. SlGSNOR shows similar K_m_ values for HMGSH and GSNO, 58 and 57 µM, respectively, while GSNO is reduced with 15–20 times higher catalytic efficiency compared to the oxidation of HMGSH [34]. Similarly, higher reaction rates of GSNO reduction compared to HMGSH oxidation were described in hGSNOR [42].

Fatty acids with medium chains (e.g., dodecanoic, decanoic, and octanoic acid), glutathione, and its derivatives (e.g., S-methylglutathione) were described as non-competitive inhibitors of plant GSNOR. Lacking an S-nitrosyl or S-hydroxymethyl group that binds to the active site zinc atom, the affinity of inhibitors GSH and S-methylglutathione is reduced by 2–3 orders of magnitude compared to GSNO and HMGSH. N6022, a pyrolle-based compound, was found to be a significantly stronger non-competitive inhibitor compared to fatty acids, inhibiting SlGSNOR at nanomolar concentrations [34].

## 4. The GSNOR Role in Plants

Biochemical and genetic characterizations of plant GSNOR enzyme, previously named either glutathione-dependent formaldehyde dehydrogenase (FALDH) or class III alcohol dehydrogenase (ADH3), have been well described in several reports [43,44,45]. Sakamoto et al. [19] identified FALDH in *Arabidopsis thaliana* as GSNOR, an enzyme able to catalyze GSNO reduction and thus regulate intracellular levels of protein S-nitrosation. GSNOR activity has been demonstrated in many plant species, e.g., *A. thaliana*, lettuce, maize, pea, rice, sunflower, and tomato [34,43,44,45,46,47,48,49,50]. Available data indicate that GSNOR is involved in numerous developmental processes and metabolic programs in plants via regulation of NO homeostasis. The enzyme is highly evolutionarily conserved [18]. Most sequenced green plant genomes encode a single copy of a GSNOR protein, predicted to be localized in cytosol [35]. The presence of multiple gene copies has only been reported in several plant species.

GSNOR is found throughout the plant suggesting the regulation of GSNO concentration in all plant cell types [51]. Experimental evidence suggests localization in the cytosol, nucleus (excluding nucleolus), and peroxisomes of *A. thaliana* [35]. Since GSNOR lacks a nuclear targeting signal, a transportation step in association with another protein is supposed. Studies on pea leaves cells showed GSNOR localized identically with *A. thaliana* in cytosol and peroxisomes and in chloroplasts and mitochondria [52]. Mitochondrial targeting peptide was predicted for *Physcomitrella* GSNOR paralog [35]. Modulation of the mitochondrial functionality by GSNOR, using cell suspension cultures with both higher and lower GSNOR levels, was demonstrated in *A. thaliana* plants [53]. Changes in GSNOR levels have an influence on the activities of mitochondrial complex I, external NADH dehydrogenase, alternative oxidase and uncoupling protein. GSNOR modulates the activity of the mitochondrial respiratory chain through controlling NO/SNO homeostasis under physiological conditions and under nutritional stress. In addition to its role in the reduction of GSNO, it may control the redox state of cells by affecting to intracellular levels of NADH and GSH.

Similarly to other organisms, plant GSNOR regulates levels of S-nitrosothiols through an irreversible NADH-dependent degradation of S-nitrosoglutathione, and it plays an important regulatory role in overall NO metabolism. Modulations of GSNOR both on the transcriptional and post-translational level can therefore contribute to a fine-tuning of NO signaling pathways in plants (Figure 2). Interestingly, reversible oxidative modification of GSNOR cysteine residues are known to inhibit its enzyme activity in vitro, suggesting a potential direct crosstalk of RNSs and ROSs signaling at this point [51,54]. Moreover, negative regulation of GSNOR activity by nitrosative modifications might present another important mechanism to control GSNO levels, a critical mediator of the downstream signaling effects of NO [38], as well as for formaldehyde detoxification in the enzyme dehydrogenase reaction mode.

### 4.1. GSNOR in Plant Growth and Development

Nitric oxide is well-known to be involved in regulation of a broad spectrum of activities during plant growth and development. Its action is supposed to be mediated via formation of S-nitrosothiols and trans-nitrosation reactions. Relative stable S-nitrosothiols enable signal transfer at large distances, S-nitrosation and denitrosation reactions are strongly controlled by the GSNOR. Although a constitutive GSNOR expression was suggested through the plant, different expression in organs of *A. thaliana* was found using histochemical activity staining and immunolocalization [30,55]. Higher levels of GSNOR were observed in the roots and leaves from the first stages of development. In transgenic *A. thaliana* plants, both up- and down regulation of GSNOR levels resulted in noticeable changes in the phenotype, namely a shortening of root length [30]. Experiments with *A. thaliana* HOT5 (sensitive to hot temperatures) mutants demonstrated that GSNOR function was necessary for normal plant growth, fertility, and plant acclimation to high temperatures [56]. Mutant plants failed to grow on nutrient plates and showed increased reproductive shoots and reduced fertility. Both *hot5* missense and null mutations showed increased NO species, supporting the statement that GSNOR regulates NO homeostasis. Furthermore, *A. thaliana* null mutants exhibit defects in stem and trichome branching [35]. The ubiquitous expression throughout the plant was confirmed using GFP-tagged GSNOR, with especially high fluorescent signal in the root tip, apical meristem, and flowers. Additional experiments [24,57,58] demonstrate that GSNOR has an influence on shoot branching, hypocotyl growth, seed yield and flowering time, decreased stature or loss of apical dominance, and fewer rosette leaves. Defective growth and development of the *gsnor1-3* mutant of *A. thaliana* with reduced GSNOR activity result from impaired, but not completely abolished, auxin signaling, auxin polar transport, and auxin distribution [58]. The processes mentioned here might be regulated by S-nitrosation of components in auxin signaling and transport, e.g., integral membrane proteins transporting auxin, intracellular receptor TIR1 (transport inhibitor response 1), and ubiquitin-conjugating enzyme E2.

Abscisic acid (ABA) is another phytohormone important for plant growth, development, and adaptation to stress conditions. ABA signaling in guard cells is impaired in *gsnor1–3* plants via S-nitrosation of sucrose nonfermenting 1 (SNF1)-related protein kinase 2.6 (SnRK2.6), which is one of the central components of the ABA signaling pathway, at cysteine 137, a residue close to kinase catalytic site [59]. Frungillo et al. [15] described the influence of GSNOR on the assimilation of nitrogen, which is a major nutrient in plant growth and development. *A. thaliana* plants overexpressing the GSNOR gene exhibit increased nitrate reductase (NR) activity; conversely, GSNOR mutant plants show a significant decrease in NR activity. Simultaneously, GSNOR enzymatic activity, but not gene expression, is inhibited by the nitrogen assimilatory pathway via post-transcriptional S-nitrosation, preventing any scavenging of GSNO. These data indicate that NO and S-nitrosothiols control their own generation and scavenging via modulation of GSNOR activity and nitrate assimilation [15]. Taken together, acquired data show that GSNOR is essential for normal growth and development of *A. thaliana*.

The spatial distribution of GSNOR activity and gene expression in pepper plants (*Capsicum annuum*) [60] was found to be in agreement with the data from *A. thaliana* [30]. At the early stages of development up to 14 days after germination, the highest activity of GSNOR was found in roots in comparison to hypocotyls and cotyledons. The activity of the enzyme decreased with age in roots and, on the contrary, increased in hypocotyls and cotyledons; however, no relevant changes in the gene expression were observed [60]. Different GSNOR gene expression was observed in organs of tomato (*Solanum lycopersicum*), with a contradictory trend during plant ageing [34]. At the early stage of development, both GSNOR gene expression and activity were found to be higher in cotyledons compared to roots, whereas the expression is higher in roots and stem compared to leaves and shoot apex at later stages. The GSNOR gene is highly expressed in stamens and pistil and in fruits during ripening. Similar to phenotypes of *A. thaliana* mutants, GSNOR overexpression in tomato plant had little effect on growth and development, whereas GSNOR downregulated plants are significantly smaller, suggesting a role for NO and S-nitrosothiol signaling [61].

### 4.2. GSNOR in Plant Responses to Abiotic Stress

Accumulated experimental evidence has delineated the importance of GSNOR in plant responses to diverse abiotic stress conditions (reviewed in [3,62]). GSNOR gene expression and enzymatic activity are altered by plant exposure to abiotic stress stimuli, e.g., low and high temperatures, wounding, continuous light and darkness and exposure to heavy metals [63,64,65,66,67,68]. Here we present an updated overview including recent advances and reports on the modulation of GSNOR gene expression and enzymatic activity by plant exposure to abiotic stress stimuli, e.g., low and high temperatures, wounding, continuous light and darkness, and exposure to heavy metals. 

#### 4.2.1. Mechanical Injury and Wounding

GSNOR gene expression is downregulated in Arabidopsis after wounding; moreover, both GSNOR mRNA and protein levels are decreased in tobacco plants after treatment with jasmonic acid, the hormone implicated in the wounding signal transduction [69]. Another study with *A. thaliana* plants described the role of GSNOR in modulating levels of GSNO and its consequence for wound response [68]. Using wild-type and GSNOR-antisense plants, the data showed wounding-induced GSNO accumulation controlled by GSNOR. The rapid increase of GSNO was observed in the injured leaves, whereas it was detected later in vascular tissues and parenchyma of systemic leaves, suggesting the role of GSNO in the wound signal transmission through vascular tissue. In addition, GSNO accumulation was required to activate the jasmonic acid-dependent wound responses at local and systemic levels [68]. GSNOR is downregulated, at the level of gene and protein expression and enzymatic activity, in mechanically damaged sunflower (*Helianthus annuus*) seedlings, which in turn leads to an accumulation of S-nitrosothiols, specifically GSNO [64]. An increase in GSNOR activity in roots, stems, and leaves was observed in two genotypes of *Cucumis* spp., *C. sativus* and *C. melo*, and pea (*Pisum sativum*) exposed to mechanical damage of stem and leaf [67]. GSNOR activity was generally higher, but any unequivocal tendency in changes in the activity in the time of experiment relevant for all studied plants was found.

A potential role of GSNOR in plant resistance to herbivory *Manduca sexta* was examined in tobacco (*N. attenuata*) plants using a virus-induced silencing of GSNOR [70]. GSNOR-silenced plants were more susceptible to herbivore attack and decreased the herbivore-induced accumulation of phytohormones jasmonic acid (JA) and ethylene and activity of trypsin proteinase inhibitors. Moreover, it was found that GSNOR mediates some jasmonate-dependent responses, e.g., the accumulation of defense secondary metabolites.

#### 4.2.2. Thermotolerance

GSNOR is involved in plant responses to cold and heat stress. Enzymatic activity of GSNOR is essential for the acclimation of *A. thaliana* plants to high temperature, since HOT5 mutants, plants with defect GSNOR gene, are more sensitive to high temperature as a consequence of disturbed homeostasis of S-nitrosothiols and NO-derived ROS signaling pathways [56]. GSNOR is expressed constitutively during plant development, and any significant regulation at the transcriptional level or at the level of protein induced by heat was observed. Posttranscriptional redox regulation, possibly by cysteine modifications, might be a mechanism by which the enzymatic activity is controlled [56]. NO and GSNO, as S-nitrosating agents, and GSNOR were found to be involved in the programmed cell death (PCD) induced by heat shock or H_2_O_2_ in tobacco (*Nicotiana tabacum*) bright yellow-2 cells [71]. NO increased in both experimentally induced PCDs, and GSNO level increased in H_2_O_2_-treated cells and decreased in cells exposed to heat shock, which is in accordance with lower GSNOR expression and activity observed in H_2_O_2_-treated cells and with higher GSNOR expression and activity in heat-shocked cells.

Low and high temperatures induce nitrosative stress in pea plants, since higher levels of NO, SNOs, and protein tyrosine nitration, markers of nitrosative stress, were detected, together with increased GSNOR activity [63]. Similarly, GSNOR activity is induced by cold stress in leaves of pepper (*Capsicum annum* L.) plants, while NO content is lower [65]. Coincidently with previous experiments, Kubienová et al. [67] described the same trend, an increase in activity of GSNOR in pea, *Cucumis sativus*, and *Cucumis melo* plants, with differences among the studied plants and their organs and in general with stronger changes induced by cold stress in comparison with heat stress. GSNOR regulates germination of recalcitrant *Baccaurea ramiflora* seeds under chilling stress probably by modulating the total RNSs content, as enzyme inhibitors dodecanoic acid and 5-chloro-3-(2-[(4-ethoxyphenyl)ethylamino]-2-oxoethyl)-1H-indole 2-carboxylicacid caused a significant increase in total RNSs and reduced germination [72]. Chilling stress enhanced the GSNOR activity and increased the level of S-nitrosothiols, while exogenous NO and CO treatment suppressed the chilling-induced accumulation of S-nitrosothiols and induced GSNOR activity. Similarly, an increase in both S-nitrosothiols and non-protein thiols was observed in plants of *Brassica juncea* under cold-stress, suggesting that S-nitrosation might regulate redox and stress-related proteins in apoplasts [73].

The role of GSNOR in the nitrosative responses was examined in citrus plants exposed to various types of abiotic stresses. GSNOR was considerably downregulated at the level of mRNA by continuous light, salinity, and especially cold, and together the enzymatic activity was decreased in plants exposed to continuous light or dark and cold [66]. A possible role for GSNOR in regulating of cytosolic redox status and SNOs content during chilling stress was also suggested in poplar (*Populus trichocarpa*), a fast growing woody plant. NO and SNO content as well as GSNOR protein and enzymatic activity were increased in poplar leaves after chilling treatment [74].

#### 4.2.3. Toxic Metals

Heavy metals have a toxic effect on plants, e.g., an induction of oxidative and nitrosative stress, leading to a severe growth inhibition, decreased photosynthesis, transpiration and chlorophyll content. The relation between ROSs and RNSs and the role of NO and enzymes affecting the NO level were examined in several plant species. Peto et al. [75] described the behavior in wild-type *Nox1* and *Gsnor1-3* mutant *A. thaliana* plants during copper stress. *Nox1* is an NO overproducing plant with higher levels of L-arginine and L-citrulline, and *Gsnor1-3* is a plant with reduced GSNOR activity with higher levels of NO, S-nitrosothiols, and nitrate [24,55,56]. The strength of the stress determines the role of NO [75]. A high NO level, due to the reduced GSNOR activity, increases sensitivity under mild stress conditions, while it supports tolerance under severe stress conditions. A forty percent increase in GSNOR activity was observed in *A. thaliana* plants grown in the presence of 0.5 mM arsenate, accompanied with a significant reduction of GSNO content and a significant increase in NO content [76]. *Gsnor1-3* mutant *A. thaliana* plants with a high S-nitrosothiols level show an increased selenite tolerance [77].

GSNOR modulates NO-induced nitrosative stress in rice plants grown under aluminum stress, which leads to accumulation of both ROSs and RNSs. GSNOR gene expression and enzymatic activity were slightly higher and the enzymatic activity was significantly increased by NO treatment in rice plants grown under aluminum stress [78]. A fast increase in S-nitrosothiol content and a reduction of the leaf photosynthesis ratio is a result of suppressed GSNOR activity with specific inhibitors. In potato plants exposed to aluminum, GSNOR activity is not affected in roots and it is increased by about 20 and 45% in leaves and stems, respectively [79]. A contrary trend in the regulation of GSNOR during heavy metal stress was observed in pea leaves treated with 50 µM cadmium, where GSNOR expression and activity were decreased by about 30% [46].

#### 4.2.4. Soil Salinity and Alkalinity

Salinity and alkalinity of soil are significant factors limiting plant growth, where NaHCO_3_ and Na_2_CO_3_ are the main contributors, leading to the creation of osmotic stress, high soil pH, and excess Na^+^. Metabolic regulation of NO and S-nitrosothiols was examined in tomato plants grown under alkaline stress [61]. GSNOR expression as well as protein is significantly inhibited in response to alkaline stress with levels fluctuating during the alkaline treatment. Plants overexpressing GSNOR are alkaline-tolerant, while under-expressing plants are alkaline-sensitive. During the alkaline treatment, overexpressing plants exhibit significantly increased efficiency of ROS scavenging, while under-expressing plants accumulate both ROSs and RNSs, thus leading to oxidative stress and programmed cell death. GSNOR may regulate tolerance of tomato plant to alkaline stress, having a role in regulating redox balance [61]. Similarly, a decrease in the GSNOR enzymatic activity was observed in roots of tomato (*Solanum lycopersicum*) plants treated with 120 mM NaCl [80]. Salinity caused an overall decrease in the content of redox molecules nicotinamide adenine dinucleotide phosphate (NADPH) and reduced glutathione (GSH), in contrast to increased NO levels. Salt stress upregulates GSNOR in citrus plants, and the GSNOR function is controlled by polyamines, substances involved in plant responses to abiotic stress. Significant suppression of GSNOR gene expression and enzymatic activity by polyamines in salinized citrus plants was reported, suggesting the role of GSNOR in modulating of nitrosative signaling [81].

Actions of NO, calmodulins (CaM), and GSNOR in *A. thaliana* plants in response to salt stress were described by Zhou et al. [82]. CaM is a significant Ca^2+^ sensor protein in plants acting as signaling molecule mediating reactions against various stresses; two isoforms *AtCaM1* and *AtCaM4*, encoding the same protein, are induced by salinity. Both AtCaM1 and AtCaM4 proteins bind directly to GSNOR. The protein–protein interaction inhibits GSNOR enzymatic activity and results in an increased NO level. Moreover, AtCaM4–GSNOR interaction regulates the ion balance, so it increases plant resistance to saline stress [82].

An important role of NO, GSNOR, and S-nitrosation in response to salt stress was described in a unicellular green alga *Chlamydomonas reinhardtii* [83]. NO production via increased nitrate reductase, but not NOS-like enzyme, activity was induced by salt stress to trigger the defense response. Induction or inactivation of antioxidant enzymes and GSNOR varied in connection with the duration of salt stress. Short-term stress caused the enzymes to scavenge ROSs and RNSs and balance cellular redox status. Long-term stress inactivated them significantly by RNS-induced protein S-nitrosation, resulting in oxidative damage and reduced cell viability. Salt stress induced the accumulation of S-nitrosothiols and S-nitrosation of GSNOR, glutathione S-transferase, and ubiquitin-like protein; S-nitrosation was reduced by thioredoxin-h5 (TRXh5), while it was enhanced by GSNOR inhibitor DA, suggesting the important role of GSNOR and S-nitrosation in adaptation of *C. reinhardtii* to salt stress [83].

#### 4.2.5. Other Abiotic Stresses

GSNOR activity is modulated in response to altered light conditions, as described for the first time in *A. thaliana* HOT5 plants grown in the dark [56]. A significant increase was observed in leaves of pea plants exposed to continuous light and continuous dark [63]. A distinct trend was observed in plants of pea, *Cucumis sativus* and *Cucumis melo*, grown either in continuous darkness (etiolated plants) or transferred to a normal light regime after 7 days in the dark (de-etiolated plants) [67]. GSNOR activity in roots decreased in time in all studied plants and was not affected by different light conditions. Continuous darkness led to a significant decrease in GSNOR activity in etiolated hypocotyls, which did not recover to values of control green plants until 168 h after de-etiolation.

Water stress, a problem for plant growth and productivity, in *Lotus japonicus* leads to both oxidative and nitrosative stress. Among others, cellular NO and S-nitrosothiol content are increased, GSNOR activity is reduced, and protein tyrosine nitration is stimulated [84]. The role of GSNOR in plants of *Lamiophlomis rotata* at high altitude was described by Ma et al. [85]. The GSNOR protein level and enzymatic activity increases in connection with a rising altitude. Since GSNOR is supposed to scavenge excess RNSs, the enzyme restricts RNA damage by decomposition of RNSs.

A direct crosstalk between ROS- and NO-dependent signaling pathways was described in *A. thaliana* plants. GSNOR activity is inhibited both by H_2_O_2_ in vitro and by oxidative stress induced by paraquat treatment in vivo, which leads to enhanced S-nitrosothiol and nitrite levels. The loss of enzymatic activity is caused by the release of one Zn^2+^ per subunit, probably that one from the active center of the protein [86]. *A. thaliana* GSNOR1/HOT5 mutant was identified to be identical to the *paraquat resistant2-1* (par2-1) mutant, showing an anti-cell death phenotype, supporting the role of GSNOR in regulation of cell death in plants via modulation of intracellular level of NO [48]. A higher NO level was found in paraquat-resistant mutant plants and, in a similar way, wild-type plants treated with an NO donor were also resistant to paraquat. The protein level of GSNOR was increased by paraquat and decreased by NO donors, while the mRNA level was not influenced.

The role of GSNOR and the regulation of intracellular SNO levels were studied in an NO accumulation mutant (*nitric oxide excess1, noe1*) in rice (*Oryza sativa*) [87]. *NOE1* was identified as a rice catalase, and an increased level of H_2_O_2_ was a result of its mutation, promoting the nitrate reductase-dependent induction of NO production. The overexpression of GSNOR gene reduces intracellular S-nitrosothiol content and alleviates cell death in the leaves of *noe1* plants. GSNOR is supposed to have a role during the desiccation of seeds of recalcitrant *Antiaris toxicaria*. Desiccation induces ROS accumulation leading to oxidative stress, enhances carbonylation, and reduces the S-nitrosation of antioxidant enzymes, antioxidant enzyme activities, and the seed germination rate. Treatment with GSNOR inhibitors dodecanoic acid or 3-[1-(4-acetylphenyl)-5-phenyl-1H-pyrrol-2-yl]propanoic acid further increases the level of antioxidant enzymes S-nitrosation and reverses seed germination inhibited by desiccation [88]. Exposure to both of these GSNOR inhibitors prior to NO gas, which is a well-known inducer of seed germination, leads to enhanced S-nitrosation and the activity of antioxidant enzymes ascorbate peroxidase, dehydroascorbate reductase, and glutathione reductase [88].

### 4.3. GSNOR in Plant Responses to Biotic Stress

NO and S-nitrosylated proteins are important signal molecules activating an immune response of plants to microbial pathogens. Thus, plant defense response to the pathogen is expected to be controlled by GSNOR, which manages NO/S-nitrosothiol homeostasis (reviewed in [89]). Díaz et al. [69] demonstrated for the first time that GSNOR gene expression is transcriptionally regulated in response to signals associated with plant defense in *A. thaliana* and tobacco. The gene expression in both mentioned plant species is induced by salicylic acid (SA), a mediator of biotic stress. The following experiments with *A. thaliana* transgenic plants [55] described GSNOR as an important component of resistance protein signaling networks. Transgenic plants with decreased GSNOR gene expression, achieved by antisense strategy, show increased content of intracellular S-nitrosothiols, constitutive activation of the pathogenesis-related (PR-1) gene, and enhanced basal resistance against biotrophic pathogen *Peronospora parasitica*. Systemic acquired resistance (SAR) was demonstrated to be enhanced in plants under-expressing GSNOR and decreased in overexpressing plants; in agreement with the rate of GSNOR expression, the level of S-nitrosothiols changed in local and systemic leaves. The SAR signal transmission might be regulated by GSNOR through the vascular system, as the enzyme was found to be localized in the phloem. Taken together with the previous published data [69], downregulation of GSNOR accompanied by an increase in the level of S-nitrosothiols is suggested to result in enhanced plant immunity [55]. The hypothesis that GSNOR is a key regulator of systemic defense responses in pathogenesis was supported by another study [68], where GSNO was found to act synergistically with salicylic acid in SAR.

Rather opposing results, when GSNOR was proposed to be a positive regulator of plant immune responses suppressing pathogen growth early in the infection process, were described earlier by Feechan et al. [24] in *A. thaliana* plants exposed to diverse microbial pathogens, e.g., bacteria *Pseudomonas syringae*, powdery mildew *Blumeria graminis*, and downy mildew *Hyaloperonospora parasitica*. Basal disease resistance was strongly reduced in the absence of GSNOR, accompanied with reduced and delayed expression of SA-dependent genes, while non-host resistance was increased in *A. thaliana* mutants overexpressing GSNOR, accompanied with accelerated expression of SA-dependent genes [24]. The content of SNOs and GSNOR activity was studied in two types of sunflower (*Helianthus annuus* L.) cultivars with different sensitivity to the pathogen *Plasmopara halstedii*, susceptible and resistant ones [47]. After infection, enzymatic activity slightly increased in a susceptible cultivar, while more explicit increase was observed in a resistant cultivar. As for the level of S-nitrosothiols after infection, it was enhanced 3.5-fold and reduced 1.5-fold in susceptible and resistant cultivars, respectively. Different spatial localization of S-nitrosothiols in hypocotyls depending on the susceptibility was observed. Different GSNOR activity was found under normal conditions in leaves of two genotypes of *Cucumis* spp. varying in susceptibility to biotrophic pathogen *Golovinomyces cichoracearum* [67]. Significantly higher enzymatic activity was found in leaves of the susceptible one compared to the moderately resistant one.

GSNOR might have an important role in NO-mediated biochemical modifications that subsequently lead to the more effective defense responses of potato plants to an attack of *Phytophthora infestans*. Potato leaf treatment with SAR inducers β-aminobutyric acid (BABA) and laminarin decreased GSNOR activity and provoked accumulation of NO and ROSs. Pre-treatment with mentioned SAR inducers before inoculation with *P. infestans* by contrast increased GSNOR activity significantly, the S-nitrosothiol pool was depleted, and potato defense responses to the pathogen were enhanced, while non-inducer pre-treated plants showed unaltered enzymatic activity, a high level of S-nitrosothiols, and lower defense responses [90]. Interestingly, no significant changes in the activity of GSNOR and S-nitrosothiols levels were observed in plants of *Medicago truncatula* after infection with *Aphanomyces euteiches* [91]. Data in that study show that resistance of *M. truncatula* against *A. euteiches* is connected with NO homeostasis, which is closely related with N nutrition.

In the plant immune response, oxidoreductase TRXh5 was found to be an effective protein-SNO reductase, providing reversibility and specificity to signaling via protein-SNO [92]. The data indicate that TRXh5 and GSNOR, enzymes exhibiting similar subcellular localization, might have partially distinct groups of protein-SNO substrates, thus regulating different immune signaling pathways. Significantly enhanced transcription of TRXh and GSNOR genes was found in transgenic plants of *Nicotiana tabacum* overexpressing γ-glutamylcysteine synthetase with higher glutathione levels, which have increased tolerance to biotrophic *Pseudomonas syringae* pv. *tabaci* [93]. GSNOR, the TRXh gene, and other genes of SA-mediated pathway dependent on non-expressor of pathogenesis-related gene 1, a transcriptional coactivator, were upregulated in tobacco BY-2 cells treated with exogenous GSH. Accumulation of NPR1 was induced by GSNO together with an enhanced SA concentration and subsequent activation of pathogenesis-related genes, leading to enhanced resistance of *A. thaliana* plants to *Pseudomonas syringae* pv. *tomato* [94]. NO induced an increase in GSH, which is indispensable to SA accumulation and non-expressor of pathogenesis-related gene 1-dependent activation of defense response. Interestingly, both SA synthesis and signaling are decreased in *Nox1*, a NO-overproducing mutant of *A. thaliana* [95]. Those plants show disabled basal resistance and resistance gene-mediated protection. Moreover, using different double mutant plants *Nox1* and *atgsnor1-1* (plant overexpressing GSNOR) or *atgsnor1-3* (plant under-expressing GSNOR), the authors suggest that NO and GSNO control cellular processes in different ways via distinct or overlapping molecular targets.

Understanding of GSNOR, thioredoxin (TRX), and their roles during biosynthesis of phenylpropanoid-derived styrylpyrone polyphenols, components inhibiting tumor proliferation and reducing hypertension and various neurodegenerative disorders [96], in co-cultured *Inonotus obliquus* and *Phellinus morii* might be, in future, employed for medicinal applications. Zhao et al. [97] described the interplay between GSNOR and TRX and their regulation via S-nitrosation/denitrosation and an impact to styrylpyrone biosynthesis. S-nitrosation of the key enzymes in the phenylpropanoid biosynthesis decreases their activity, which can be restored by TRX-mediated denitrosation. Moreover, TRX acts as a trans-nitrosylase leading to the S-nitrosation of GSNOR via a protein–protein interaction and thus a decrease in its enzymatic activity.

## 5. Conclusions

S-nitrosation has emerged among the key components of redox-based NO signaling that regulate the structure and activity of proteins through reversible post-translational modification of cysteine thiols. Despite the important advances in the understanding of the functions of S-nitrosation and S-nitrosothiols in plant metabolism and stress responses, major gaps in the picture of S-nitrosation on the intersection of signaling pathways of NO and ROSs still remain. Among them, the identification of NO sources and their localization contributing to S-nitrosation reactions in distinct tissues, cells, and subcellular compartments continues to be crucial in plant NO research in general. On the other hand, the mechanisms of in vivo regulations regarding the activity of GSNOR, and potentially of other more specific denitrosylases acting directly on proteins S-nitrosothiol, are still poorly described. Moreover, the development of highly specific and sensitive analytical tools to evaluate levels of both low molecular- and protein S-nitrosothiols will certainly contribute to advancement in the plant S-nitrosation field. Finally, the transfer of the knowledge obtained with model plants such as *A. thaliana* to important agricultural crops is expected to be exploited through genetic manipulation of GSNOR levels to eventually achieve desired improvements in crop yields and stress tolerance.

## Figures and Tables

**Figure 1 plants-08-00048-f001:**
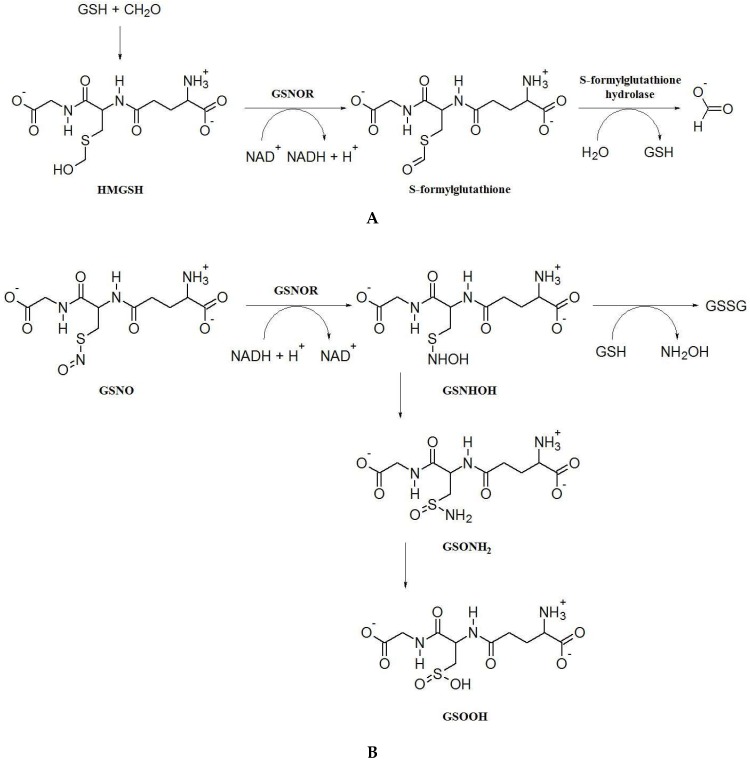
Reaction mechanisms of alcohol dehydrogenase/S-nitrosoglutathione reductase (ADH3/GSNOR) in formaldehyde and S-nitrosoglutathione catabolism. (**A**) In the dehydrogenase mode, GSNOR using NAD^+^ as a coenzyme catalyzes the oxidation of S-hydroxymethylglutathione (HMGSH), spontaneously formed from formaldehyde and glutathione to S-formylglutathione, which was further hydrolyzed to glutathione and formate by S-formylglutathione hydrolase. (**B**) In the reductase mode, GSNOR catalyzes the reduction of S-nitrosoglutathione (GSNO) using NADH to an unstable intermediate N-hydroxysulfinamide (GSNHOH). Depending on the local concentration of GSH, GSNHOH is either decomposed to glutathione disulfide (GSSG) and hydroxylamine at high GSH levels, or at low GSH levels spontaneously converts to glutathione sulfinamid (GSONH2), which can be hydrolyzed to glutathione sulfinic acid (GSOOH) and ammonia.

**Figure 2 plants-08-00048-f002:**
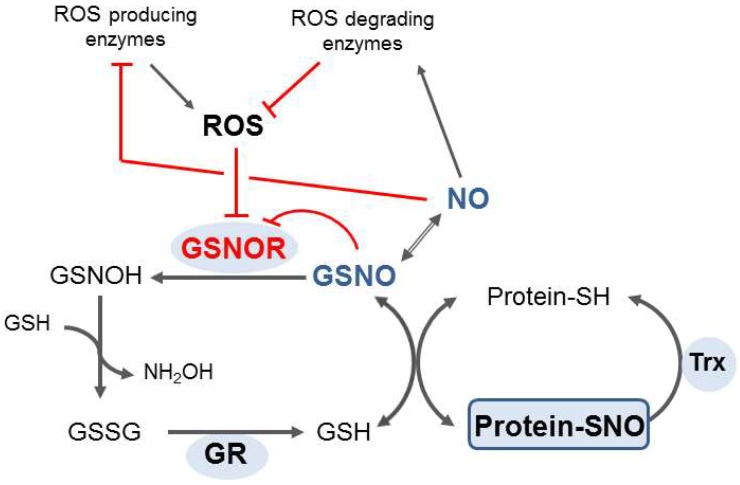
Regulatory mechanisms of GSNOR in protein denitrosation on the intersection of signaling pathways of ROSs and RNSs. Trans-nitrosation reactions of S-nitrosated proteins and reduced glutathione (GSH) can be shifted by the GSNOR activity through irreversible NADH-dependent reduction of S-nitrosoglutathione. GSH can be eventually regenerated by an NADPH-dependent reduction of GSSG catalyzed by glutathione reductase (GR). GSNOR activity can be inhibited by oxidative modification, resulting in GSNO accumulation and hence increased NO bioactivity, which can in turn regulate activities of enzymes of ROS metabolism. GSNOR activity can be also inhibited by S-nitrosation, to enable transient accumulations of its substrate GSNO and eventually to influence the cellular status of protein S-nitrosation.

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
