# Peer review of "S-Nitrosoglutathione Reductase—The Master Regulator of Protein S-Nitrosation in Plant NO Signaling"

_plants, 2019, doi:10.3390/plants8020048_

Round 1
Reviewer 1 Report
This paper by Jahnova et al. presents current knowledge on the GSNO reductase and its role in plant physiology. GSNOR activity is mainly described through its interaction with NO signaling.
In the first part of the review, the authors introduce some basic concepts concerning GSNO reductase activity and molecular features. The second, and more significant part, concerns the GSNOR role in plant physiology (growth/development and responses to stress).
In general, the paper is well-written and the ideas are well organized. It provides a thorough contribution to the understanding of this enzyme, whose role is highly investigated in plant since several years.
I have nevertheless several (major and minor) comments :
- The main comment concerns the description of the link/ correlation between GSNOR activity and SNO concentration (or NO accumulation). This crucial point should be better promoted (in accordance with the title of the paper). Indeed the effect of GSNOR activity on SNO or NO “concentrations” and the way it could control these molecules appear sometimes contradictory in the literature (depending stress conditions, plant species…). More emphasis on GSNOR regulation (by redox state, nitrosation…) could be also interesting. In this way, and to facilitate the reading and improve the value of the review, I suggest the authors to add figure(s) illustrating the role of regulator of GSNOR (link with NO signalling), its regulation and to present the general trends of its role in plant physiology (or to write a specific paragraph summarizing these points). Two recent papers on that topic could be also quoted : Tichá et al. "Redox regulation of plant S-nitrosoglutathione reductase activity through post-translational modifications of cysteine residues." Biochemical and biophysical research communications 494.1-2 (2017): 27-33.
Lindermayr, C. (2018). Crosstalk between reactive oxygen species and nitric oxide in plants: key role of S-nitrosoglutathione reductase. Free Radical Biology and Medicine, 122, 110-115.
- Concerning the introduction: I know that it is difficult to be representative of the huge amount of papers concerning NO in plants but a reference concerning role of NR could be added (e.g. Sakihama et al. 2002) as well as the evolutionary analysis of NOS indicating the absence of NOS gene in land plants (Jeandroz et al. 2016). See below for complete ref.
- Concerning paragraph 4.2 (GSNOR in plant response to abiotic stress.) Line 272 : the authors should justify the originality of their paper(compared to the 2 other cited reviews [3]and [62]).
- Line 382 : Chlamydomonas reinhardtii is not situated between animals and higher plants. Cr belongs to Chlorophyte a sister group of land plants, and to viriplantae together with land plants.
- Conclusion: line 519 the manipulation of GSNOR in GMO will be, in my opinion, difficult as the modulation of GSNOR activity could detrimental or beneficial for plants.
- Finally, it contains some minor typing problems that should be edited
line 311 “treated” instead of “tated”
line 540 : 565 (protein)??
Sakihama, et al.. "Nitric oxide production mediated by nitrate reductase in the green alga Chlamydomonas reinhardtii: an alternative NO production pathway in photosynthetic organisms." Plant and Cell Physiology 43.3 (2002): 290-297.
Jeandroz et al. "Occurrence, structure, and evolution of nitric oxide synthase–like proteins in the plant kingdom." Sci. Signal. 9.417 (2016): re2-re2.
Author Response
Reviewer 1
This paper by Jahnova et al. presents current knowledge on the GSNO reductase and its role in plant physiology. GSNOR activity is mainly described through its interaction with NO signaling. In the first part of the review, the authors introduce some basic concepts concerning GSNO reductase activity and molecular features. The second, and more significant part, concerns the GSNOR role in plant physiology (growth/development and responses to stress).
In general, the paper is well-written and the ideas are well organized. It provides a thorough contribution to the understanding of this enzyme, whose role is highly investigated in plant since several years. I have nevertheless several (major and minor) comments :
Point 1 - The main comment concerns the description of the link/ correlation between GSNOR activity and SNO concentration (or NO accumulation). This crucial point should be better promoted (in accordance with the title of the paper). Indeed the effect of GSNOR activity on SNO or NO “concentrations” and the way it could control these molecules appear sometimes contradictory in the literature (depending stress conditions, plant species…). More emphasis on GSNOR regulation (by redox state, nitrosation…) could be also interesting. In this way, and to facilitate the reading and improve the value of the review, I suggest the authors to add figure(s) illustrating the role of regulator of GSNOR (link with NO signalling), its regulation and to present the general trends of its role in plant physiology (or to write a specific paragraph summarizing these points). Two recent papers on that topic could be also quoted:
Tichá et al. "Redox regulation of plant S-nitrosoglutathione reductase activity through post-translational modifications of cysteine residues." Biochemical and biophysical research communications 494.1-2 (2017): 27-33.
Lindermayr, C. (2018). Crosstalk between reactive oxygen species and nitric oxide in plants: key role of S-nitrosoglutathione reductase. Free Radical Biology and Medicine, 122, 110-115.
Response 1: We have added a new paragraph in the section “GSNOR role in plants” describing the proposed role of GSNOR in the regulation of NO and ROS signaling, including suggested citations. We have also introduced two figures to illustrate the mechanisms of GSNOR reactions and regulations described in the text.
Point 2 - Concerning the introduction: I know that it is difficult to be representative of the huge amount of papers concerning NO in plants but a reference concerning role of NR could be added (e.g. Sakihama et al. 2002) as well as the evolutionary analysis of NOS indicating the absence of NOS gene in land plants (Jeandroz et al. 2016). See below for complete ref.
Response 2: We appreciate these suggestions and we have included citations of the mentioned references to the Introduction part.
Sakihama, et al.. "Nitric oxide production mediated by nitrate reductase in the green alga Chlamydomonas reinhardtii: an alternative NO production pathway in photosynthetic organisms." Plant and Cell Physiology 43.3 (2002): 290-297.
Jeandroz et al. "Occurrence, structure, and evolution of nitric oxide synthase–like proteins in the plant kingdom." Sci. Signal. 9.417 (2016): re2-re2.
Point 3 - Concerning paragraph 4.2 (GSNOR in plant response to abiotic stress.) Line 272 : the authors should justify the originality of their paper(compared to the 2 other cited reviews [3]and [62]).
Response 3: We have adjusted the introductory sentence of the paragraph 4.2. to stress the originality of our paper in respect to previously published reviews.
Point 4 - Line 382: Chlamydomonas reinhardtii is not situated between animals and higher plants. Cr belongs to Chlorophyte a sister group of land plants, and to viriplantae together with land plants.
Response 4: In this sentence, we used the information given by the authors of the cited reference by Chen et al. (2016). We have adjusted this sentence as suggested by the reviewer.
Point 5 - Conclusion: line 519 the manipulation of GSNOR in GMO will be, in my opinion, difficult as the modulation of GSNOR activity could detrimental or beneficial for plants.
Response 5: We agree that this will be a rather difficult task, nevertheless we believe that compartment -, tissue- or developmental stage-specific and controlled modulation of GSNOR activity might be exploited namely in improving crop resistance to stress conditions.
Point 6 - Finally, it contains some minor typing problems that should be edited
line 311 “treated” instead of “tated”
line 540 : 565 (protein)??
Response 6: We have corrected these typing errors.
Reviewer 2 Report
The review by Jahnová et al. presents an overview of GSNOR covering a wide range of studies, and in relation to this, two concerns are brought out for the authors to address.
1. Given the complexity of the reactions and regulations GSNOR is involved, a schematic model should be included, either under the section “S-nitrosoglutathione reductase: key enzyme of the regulation of S-nitrosation and formaldehyde detoxification”, or as a whole linking this section and “GSNOR role in plants” together. See these articles for reference.
1) S-nitrosoglutathione reductase affords protection against pathogens in Arabidopsis, both locally and systemically; 2) Nitro-oxidative stress vs oxidative or nitrosative stress in higher plants; 3) Current overview of S-nitrosoglutathione (GSNO) in higher plants.
2. It is understandable focusing on GSNOR in this review, especially when it comes to the biological functions with the versality of nitric oxide (NO) centering in many processes.
1) Under “GSNOR role in plants”, it would be beneficial to have the content from line 196 to line 220 revised in a way favors NO homeostasis and functions, and most importantly, and acknowledges the “friend or foe” nature of NO, which may operate distinctly even in the same process/species, as this was in line with the GSNOR roles as shown in the examples, and thus, 2) some of the opening or concluding sentences throughout the “GSNOR role in plants” section shall be adjusted towards this end.
Author Response
Reviewer 2
The review by Jahnová et al. presents an overview of GSNOR covering a wide range of studies, and in relation to this, two concerns are brought out for the authors to address.
Point 1 . Given the complexity of the reactions and regulations GSNOR is involved, a schematic model should be included, either under the section “S-nitrosoglutathione reductase: key enzyme of the regulation of S-nitrosation and formaldehyde detoxification”, or as a whole linking this section and “GSNOR role in plants” together. See these articles for reference.
1) S-nitrosoglutathione reductase affords protection against pathogens in Arabidopsis, both locally and systemically; 2) Nitro-oxidative stress vs oxidative or nitrosative stress in higher plants; 3) Current overview of S-nitrosoglutathione (GSNO) in higher plants.
Response 1: We agree with this reviewer suggestion and we have included a scheme depicting the GSNOR functions in plants (see new Figure 1).
Point 2. It is understandable focusing on GSNOR in this review, especially when it comes to the biological functions with the versality of nitric oxide (NO) centering in many processes.
1) Under “GSNOR role in plants”, it would be beneficial to have the content from line 196 to line 220 revised in a way favors NO homeostasis and functions, and most importantly, and acknowledges the “friend or foe” nature of NO, which may operate distinctly even in the same process/species, as this was in line with the GSNOR roles as shown in the examples, and thus, 2) some of the opening or concluding sentences throughout the “GSNOR role in plants” section shall be adjusted towards this end.
Response 2: As suggested by the reviewer, to reflect more accurately GSNOR roles in NO homeostasis, with adjusted several sentences in the introduction and “GSNOR role in plants”,, using suggested literature citations; and we have also complemented the section of “GSNOR role in plants” with one new paragraph.